# MGMapNet: Multi-granularity Representation Learning for End-to-end Vectorized HD Map Construction

**Jing Yang** [1*]     **Minyue Jiang** [2*]     **Sen Yang** [2*]     **Xiao Tan** [2]
**Yingying Li** [2]     **Errui Ding** [2]     **Jingdong Wang** [2]     **Hanli Wang** [1†]

[1] College of Electronic and Information Engineering, Tongji University, China     [2] Baidu Inc.

## Abstract

The construction of vectorized high-definition map typically requires capturing both category and geometry information of map elements. Current state-of-the-art methods often adopt solely either point-level or instance-level representation, overlooking the strong intrinsic relationship between points and instances. In this work, we propose a simple yet efficient framework named MGMapNet (multi-granularity map network) to model map elements with multi-granularity representation, integrating both coarse-grained instance-level and fine-grained point-level queries. Specifically, these two granularities of queries are generated from the multi-scale bird's eye view features using a proposed multi-granularity aggregator. In this module, instance-level query aggregates features over the entire scope covered by an instance, and the point-level query aggregates features locally. Furthermore, a point-instance interaction module is designed to encourage information exchange between instance-level and point-level queries. Experimental results demonstrate that the proposed MGMapNet achieves state-of-the-art performances, surpassing MapTRv2 by 5.3 mAP on the nuScenes dataset and 4.4 mAP on the Argoverse2 dataset, respectively.

## 1 Introduction

It is crucial to perceive and understand road map elements for ensuring the safety in autonomous driving applications (Xiao et al., 2020; Xu et al., 2023; Prakash et al., 2021). High-definition (HD) maps provide category and geometry information about road elements, enabling autonomous vehicles to maintain lane position, anticipate intersections, and plan optimal routes to mitigate potential risks. However, constructing HD maps requires significant human effort for annotating and updating, which limits scalability over large areas. Recent researches, such as (Li et al., 2022a; Liao et al., 2022; 2023; Ding et al., 2023; Yuan et al., 2024; Hu et al., 2021), focus on learning-based methods as alternatives to construct HD maps from onboard sensors. These methods can be mainly divided into two categories based on the representation in use: rasterized map-based representation (Li et al., 2022a;b; Liu et al., 2023b; Xiong et al., 2023) and vectorized map-based representation (Ding et al., 2023; Li et al., 2023; Liao et al., 2023).

Rasterized map-based methods often require complex post-processing to meet the need of downstream modules, such as planning, and suboptimal results are usually obtained which are not entirely end-to-end optimized. Therefore, there has been increasing attention paid to end-to-end map construction methods (Shin et al., 2023; Qiao et al., 2023b; Zhang et al., 2024) using vectorized representations, which commonly employ bird's eye view (BEV) (Fadadu et al., 2022; Chen et al., 2017; Liang et al., 2019; You et al., 2019; Liang et al., 2018) space for end-to-end perception, effectively integrating various sensor information such as surround-view cameras and Lidar.

The state-of-the-art methods typically adopt DETR-like architectures (Carion et al., 2020), comprising encoder and decoder components. The encoder extracts multi-sensor information into BEV

---

*Equal Contribution.

†Corresponding author (H. Wang, E-mail: hanliwang@tongji.edu.cn).

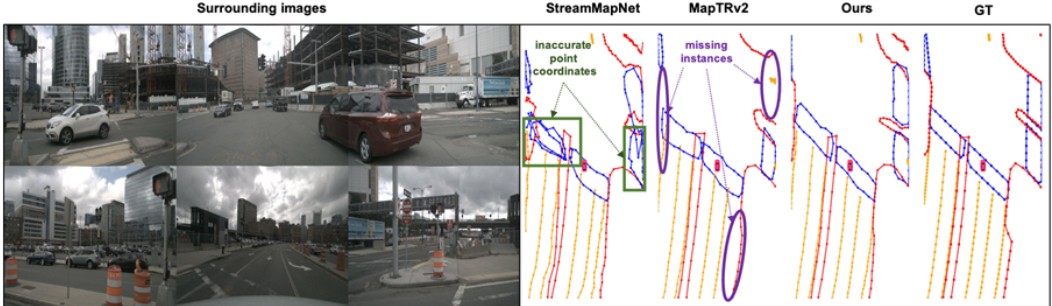

Figure 1: **Comparison of visualization results.** Visual comparison among StreamMapNet, MapTRv2, and MGMapNet, with vehicle-centric views featuring red boundaries, orange dividers, and blue pedestrian crossings. The green boxes denote the phenomenon of inaccurate point coordinates in instance-level queries, while the purple ellipses indicate the phenomenon of missing instances in point-level queries. StreamMapNet employs MPA for single-frame results. Best viewd in color.

representation, while the decoder decodes the category and geometry information of each road element through queries, and thus an end-to-end vectorized representation of output map elements is achieved, eliminating the need for the complex post-processing steps involved in rasterized map representation. These methods use either point-level queries or instance-level queries to generate map elements. Point-level queries are good at describing the geometric position of road elements. For instance, in MapTR (Liao et al., 2022) and MapTRv2 (Liao et al., 2023), a permutation-equivalent point expression accurately represents the location of map elements, ensuring stable training processes. However, these methods may lack an overall description of map elements, leading to deficiencies in representing lane relationships. For example, MapTRv2 may miss lane lines in distant and merging scenarios, as the region illustrated in the purple ellipses of Fig. 1.

While instance-level queries excel at capturing the overall category information of road elements, they may struggle to accurately represent geometric details, especially for irregular or elongated map elements. For example, in StreamMapNet (Yuan et al., 2024), multi-point attention (MPA) is proposed to capture the overall information of road elements, allowing for longer attention ranges while maintaining computational efficiency. However, this method may encounter difficulties in perceiving the geometry of irregular or elongated elements, leading to local disturbances. The green boxes in Fig. 1 highlight the issue of inaccurate point coordinates obtained from instance-level queries: although map elements are successfully detected, their positional accuracy is compromised.

The primary challenge lies in balancing detailed and comprehensive representations, which current researches and methods fail to adequately address. To integrate both fine-grained local positions and coarse-grained global classification information, we propose a multi-granularity map network (MGMapNet) to represent map elements using multi-granularity queries. Within each decoder layer, point-level queries and instance-level queries are simultaneously computed by querying multi-scale BEV features using multi-granularity aggregator. Subsequently, point-instance interaction, including point-to-point attention and point-to-instance attention, is designed to enhance intrinsic relationships. Ultimately, point-granularity queries are utilized to localize point coordinates, while instance-granularity queries are employed to determine the categories of map elements.

The main contributions of this work are summarized as follows. First, we propose a robust multi-granularity representation, enabling the end-to-end construction of vectorized HD maps by employing coarse-grained instance-level and fine-grained point-level queries in one framework. Second, the multi-granularity aggregator, combined with point-instance interaction, facilitates an efficient interaction between point-level and instance-level queries, effectively exchanging category and geometry information. Third, we incorporate several strategy optimizations into model training, enabling our proposed MGMapNet to achieve state-of-the-art single-frame performances on both the nuScenes (Caesar et al., 2020) and Argoverse2 (Wilson et al., 2023) datasets.

## 2 RELATED WORK

**Online HD Map Construction.** In recent years, researchers have utilized onboard sensors in autonomous driving to construct HD maps. The previous works (Huang et al., 2023; Chen et al., 2022)

focus on projecting and lifting map elements detected on the perspective view plane into 3D space for map reconstruction. With the aim of integrating multiple sensors such as panoramic cameras and LiDAR, construction methods for online HD map are gradually transitioning to BEV representation. The HD map construction methods can be broadly categorized into two types: rasterized map-based and vectorized map-based. The rasterized map-based methods, such as HDMapNet (Li et al., 2022a), utilize BEV features for semantic segmentation, followed by a post-processing step to obtain vectorized map instances. Similarly, BEV-LaneDet (Wang et al., 2023) introduces an efficient key-point representation for 3D lanes, generating confidence scores, y-axis offsets, and heights for each BEV grid cell. While rasterized maps can provide detailed road information, the requirement of post-processing limits their applications. With the emergence of vectorized DETR-like (Carion et al., 2020) end-to-end methods, the need for post-processing is eliminated. VectorMapNet (Liu et al., 2023a) is the first end-to-end map reconstruction model that utilizes transformers. MapTR and MapTRv2 (Liao et al., 2022; 2023) introduce novel and unified modeling methods for map elements, addressing ambiguity and ensuring stable learning processes. PivotNet (Ding et al., 2023) leverages pivot-based representations for map elements, organizing element point sets into pivotal and collinear point sequences to achieve more precise map element learning.

However, these methods often exclusively use either point-level queries or instance-level queries, missing out on the mutual advantages of both granularities. To address this limitation, we introduce a multi-granularity mechanism to represent map elements, which adaptively derives features at both fine-grained point granularity and coarse-grained instance granularity, thus preserving local details as well as global map information.

**Lane Detection.** Lane detection can be regarded as a subtask of HD map construction, focusing on the detection of lane elements within road scenes. Current methods (Li et al., 2019; Zheng et al., 2022; Tabelini et al., 2021b) predominantly engage in lane detection from a single perspective view image, and the majority of lane detection datasets provide annotations only from a single perspective. LaneATT (Tabelini et al., 2021a) proposes an anchor-based attention mechanism to aggregate global information. Unlike lane detection, vectorized HD map construction involves more complex map elements within the vehicle's perception range, including lane markings, curbs, and sidewalks.

## 3 METHOD

### 3.1 OVERALL ARCHITECTURE

The overall architecture of MGMapNet is depicted in Fig. 2 (a). Similar to other DETR-like end-to-end HD map construction models, MGMapNet comprises a BEV feature encoder which is responsible to extract multi-scale BEV features from perspective view images, and a Transformer Decoder which stacks multiple layers of multi-granularity attention (MGA) to generate predictions for map elements. The prediction from each layer encapsulates both category and geometry information within the perception range.

**BEV Feature Encoder.** The proposed MGMapNet takes surrounding-view RGB images as inputs, expressing them as unified perceptual BEV feature representation for subsequent Transformer decoder. The unified BEV feature is denoted as $\mathbf{F}_{bev} \in \mathbb{R}^{C \times H \times W}$, where $C, H, W$ represent the number of feature channels, height, and width of the BEV feature, respectively. Given the diverse lengths of map elements, relying solely on a single-scale BEV feature fails to meet the requirements for detecting all elements with different lengths. Therefore, we employ downsample modules to reduce the spatial resolution of BEV features $\mathbf{F}_{bev}$ by half, generating $\mathbf{F}'_{bev} \in \mathbb{R}^{C \times \frac{H}{2} \times \frac{W}{2}}$. More scales might be benificial, but it is observed that two scales are already good enough. As a result, $\mathbf{F}_{ms\_bev} \in \mathbb{R}^{C \times (\frac{H}{2} \times \frac{W}{2} + H \times W)}$ represents multi-scale BEV features, which are obtained by concatenating the flattened tensors of $\mathbf{F}_{bev}$ and $\mathbf{F}'_{bev}$.

**Decoder.** Figure 2 (b) illustrates the $l$-th MGA decoder layer, which is composed of self attention, MGA, and feed-forward network. The MGA consists of two components: multi-granularity aggregator and point-instance interaction. The instance-level query is initialized using learnable parameters and updated through interaction with BEV features, while the point query is dynamically generated by aggregating BEV features. Subsequently, the point-instance interaction facilitates the

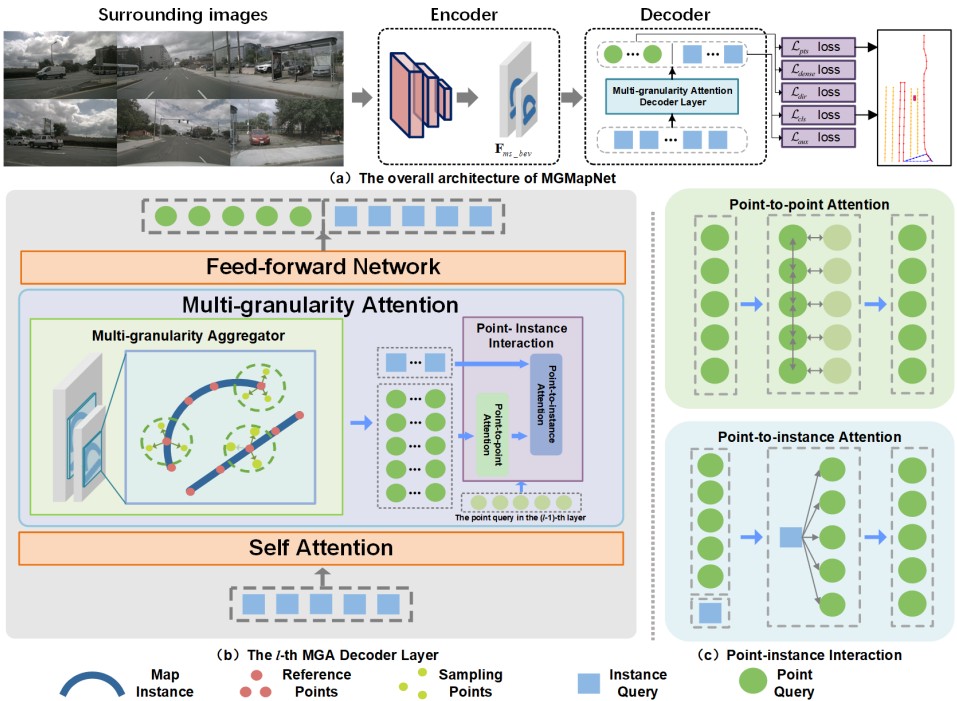

Figure 2: Overview of MGMapNet. (a) MGMapNet takes multi-view images as inputs, processes them through an encoder-decoder framework, and generates vectorized map representations. (b) The schematic diagram of the $l$-th multi-granularity attention (MGA) decoder layer. (c) Implementation details of point-instance interaction, comprising point-to-point attention and point-to-instance attention.

mutual interaction among local geometric information, global category information, and the point queries from the $(l-1)$-th layer.

## 3.2 MULTI-GRANULARITY ATTENTION

Instance-level queries effectively capture the overall categorical information of road elements but may struggle to represent geometric details, particularly for irregularly shaped or elongated map features. Conversely, point-level queries provide detailed information, however, they only represent instances by aggregating multiple queries, resulting in a lack of comprehensive descriptions of map elements. To simultaneously capture both detailed and comprehensive instance features, the multi-granularity attention (MGA) mechanism is designed to effectively maintain and update queries at various granularities. As illustrated in Fig. 2, MGA comprises two primary components: multi-granularity aggregator and point-instance interaction.

### 3.2.1 MULTI-GRANULARITY AGGREGATOR

In multi-granularity aggregator, instance-level queries interact with the multi-scale BEV features, and point-level queries are generated. Specifically, we improve the multi-head deformable attention (Zhu et al., 2020) with multiple reference points for each query to aggregate long-range features from multi-scale BEV features. To improve readability, we omit the subscript $m$ for the index of multiple heads $M$ in the operator. More specifically, the multi-granularity aggregator takes as input the instance-level queries $\mathbf{Q}_{ins} \in \mathbb{R}^{N_q \times C}$ in the first layer, along with the point-level queries $\mathbf{Q}_{pts} \in \mathbb{R}^{N_q \times N_p \times C}$ and the reference points $\mathbf{RF} \in \mathbb{R}^{N_q \times N_p \times 2}$ in the subsequent layers. $N_q$ is the total number of instance-level queries, and $N_p$ is the total number of points belonging to an instance. Noted that the reference points in the first layer are predicted by $\mathbf{Q}_{ins}$, and the reference points in

subsequent layers are updated by reference points from the previous layer as

$$\begin{cases} \mathbf{RF}^l = \mathrm{MLP}(\mathbf{Q}^l_{ins}), l = 0, \\ \mathbf{RF}^l = \mathrm{sigmoid}(\mathrm{sigmoid}^{-1}(\mathbf{RF}^{l-1}) + \mathrm{MLP}(\mathbf{Q}^l_{pts})), l >= 1, \end{cases} \tag{1}$$

where $l$ represents the layer index, $\mathrm{sigmoid}(\cdot)$ and $\mathrm{sigmoid}^{-1}(\cdot)$ refer to the sigmoid and inverse sigmoid activation functions, and $\mathrm{MLP}(\cdot)$ stands for multi-layer perceptron.

Since an instance is represented as a point sequence, position encoding is added to the instance-level query. Given the location of reference point $\mathbf{RF}$, we employ $\mathbf{RF}$ to generate positional encoding $\mathbf{PE}_{ref}$ as

$$\mathbf{PE}^{l-1}_{ref} = \mathrm{MLP}^{l-1}_{ref}(\mathbf{RF}^{l-1}), \tag{2}$$

where $\mathrm{MLP}^{l-1}_{ref}$ is a projection layer used to generate the positional embedding from reference points. We allocate $N_{rep}$ sampling points to each reference point, aggregating features from these points to enhance the representation of the reference point. The location offset $\Delta\mathbf{S}$ of sampling points w.r.t the reference point and the associated weights $\mathbf{W}$ are computed by combining the instance-level queries $\mathbf{Q}_{ins}$ and $\mathbf{PE}_{ref}$ as

$$\Delta\mathbf{S}^l = \mathrm{Sampling\_Offset}(\mathbf{Q}^{l-1}_{ins} + \mathbf{PE}^{l-1}_{ref}) \in \mathbf{R}^{N_q \times N_p \times N_{rep} \times 2},$$
$$\mathbf{W}^l = \mathrm{Weight\_Embed}(\mathbf{Q}^{l-1}_{ins} + \mathbf{PE}^{l-1}_{ref}) \in \mathbb{R}^{N_q \times N_p \times N_{rep}}, \tag{3}$$
$$\mathbf{S}^l = (\mathbf{RF}^{l-1} + \Delta\mathbf{S}^l) \in \mathbb{R}^{N_q \times N_p \times N_{rep} \times 2},$$

where $\mathrm{Sampling\_Offset}(\cdot)$ and $\mathrm{Weight\_Embed}(\cdot)$ are MLP layers designed to generate the location offset $\Delta\mathbf{S}$ and the attention weight $\mathbf{W}$, respectively; $\mathbf{RF}^{l-1}$ is expanded appropriately to match the dimension of $\Delta\mathbf{S}^l$. By leveraging the sampling offset and the reference point, the sampling location $\mathbf{S}^l$ is updated by adding $\mathbf{RF}^{l-1}$ and $\Delta\mathbf{S}^l$. Subsequently, $\mathbf{Q}_{ins}$ and $\mathbf{Q}_{pts}$ are generated by the weighted sum of sampled features as

$$\mathbf{W}^l_{ins} = \underset{(j,k)\in(N_p, N_{rep})}{\mathrm{softmax}} \left( \mathbf{W}^l_{j,k} \right) \in \mathbb{R}^{N_q \times (N_p \times N_{rep})},$$
$$\mathbf{W}^l_{pts} = \underset{k\in N_{rep}}{\mathrm{softmax}} \left( \mathbf{W}^l_{j,k} \right) \in \mathbb{R}^{N_q \times N_p \times N_{rep}},$$
$$\mathbf{Q}^l_{ins} = \sum_{j=1}^{N_p} \sum_{k=1}^{N_{rep}} \left[ \mathbf{W}^l_{ins} \mathrm{sampling}(\mathbf{F}_{\mathrm{ms\_bev}}, \mathbf{S}^l_{j,k}) \right] \in \mathbb{R}^{N_q \times C}, \tag{4}$$
$$\mathbf{Q}^l_{pts} = \sum_{k=1}^{N_{rep}} \left[ \mathbf{W}^l_{pts} \mathrm{sampling}(\mathbf{F}_{\mathrm{ms\_bev}}, \mathbf{S}^l_{j,k}) \right] \in \mathbb{R}^{N_q \times N_p \times C},$$

where $j$ is the index of the $N_p$ points on an instance, $k$ is the index among the $N_{rep}$ sampling points assigned to the reference point, $\mathbf{W}^l_{ins}$ and $\mathbf{W}^l_{pts}$ denote the softmax-normalized weights across $N_p \times N_{rep}$ and $N_{rep}$ dimensions of $\mathbf{W}^l_{j,k}$, while $\mathrm{sampling}(\cdot)$ represents the bilinear sampling operator.

Through the multi-granularity aggregator, $\mathbf{Q}_{ins}$ and $\mathbf{Q}_{pts}$ are generated from multi-scale BEV features, capturing both global and local information for each map element. Compared with the multi-point attention in StreamMapNet (Yuan et al., 2024), our method incorporates point-level queries directly from multi-scale BEV features by sampling points and enhances the accuracy of predicted geometry points. In addition, compared with point-level representations such as MapTR (Liao et al., 2022) and MapTRv2 (Liao et al., 2023), our method updates instance-level queries with sampled point features and captures the overall category as well as shape information of road elements.

### 3.2.2 POINT-INSTANCE INTERACTION

The point-instance interaction is designed with the intention of enhancing positional and categorical information interaction between these two granularities of queries. As illustrated in Fig. 2(c), point-instance interaction comprises two attention operators: point-to-point (P2P) attention and point-to-instance (P2I) attention. Concurrently, the sampling locations $\mathbf{S}^l$ and the attention weights

$\mathbf{W}_{ins}^l, \mathbf{W}_{pts}^l$ obtained from the multi-granularity aggregator in the $l$-th layer are flattened and concatenated to encode positional information in P2P attention and P2I attention as

$$\begin{aligned}
\mathbf{PE}_{ins}^l &= \mathrm{MLP}_{ins}^l(\mathbf{S}^l, \mathbf{W}_{ins}^l), \\
\mathbf{PE}_{pts}^l &= \mathrm{MLP}_{pts}^l(\mathbf{S}^l, \mathbf{W}_{pts}^l),
\end{aligned} \tag{5}$$

where $\mathrm{MLP}_{ins}^l(\cdot)$ and $\mathrm{MLP}_{pts}^l(\cdot)$ are MLP layers for instance-level queries and point-level queries respectively, $\mathbf{PE}_{ins}$ and $\mathbf{PE}_{pts}$ stand for the corresponding generated position embedding.

**P2P Attention.** As the coordinates of map elements are refined based on the point-level queries in previous MGA layer, these point-level queries play a pivotal role in predicting coordinates in the current layer. Hence, the P2P attention module is devised to include point-level queries from both the current $l$-th layer and previous $(l-1)$-th layer as inputs of the attention layer as

$$\begin{cases}
\mathbf{Q}_{pts}^{l'} = \mathrm{SA}(\mathbf{Q}_{pts}^l + \mathbf{PE}_{pts}^l), l = 0, \\
\mathbf{Q}_{pts}^{l'} = \mathrm{CA}(\mathbf{Q}_{pts}^l + \mathbf{PE}_{pts}^l, \mathbf{Q}_{pts}^{l-1} + \mathbf{PE}_{pts}^{l-1}), l >= 1.
\end{cases} \tag{6}$$

Regarding the first MGA layer, since there is not previous decoder layer before it, the self attention operation $\mathrm{SA}(\cdot)$ only conducts with the current point-level queries. For subsequent MGA layers, P2P attention is implemented through the cross attention operation $\mathrm{CA}(\cdot)$ by combining the point-level queries $\mathbf{Q}_{pts}^{l-1}$ of the previous layer with the point-level queries $\mathbf{Q}_{pts}^l$ of the current layer.

**P2I Attention.** After P2P attention, P2I attention updates the point-level queries through cross-attention to capture instance-level geometric information, while leveraging position embeddings $\mathbf{PE}_{ins}$ and $\mathbf{PE}_{pts}$ to focus on the spatial distribution of polylines as

$$\mathbf{Q}_{pts}^{l''} = \mathrm{CA}(\mathbf{Q}_{pts}^{l'} + \mathbf{PE}_{pts}^l, \mathbf{Q}_{ins}^l + \mathbf{PE}_{ins}^l). \tag{7}$$

The $N_p$ point-level queries belonging to the same instance-level query are aggregated through the aggregation layer $\mathrm{MLP}_{agg}$, producing instance-level queries as

$$\mathbf{Q}_{ins}^{l'} = \mathrm{MLP}_{agg}(\sum_{j=1}^{N_p} \mathbf{Q}_{pts,j}^{l''}). \tag{8}$$

**Output.** Ultimately, point-level queries are utilized to predict point location using a regression head, while instance-level queries are employed to predict the categories of map elements using a classification head. In summary, multi-granularity aggregator and point-instance interaction are applied to generate and update multi-granularity queries, enabling perception of both the geometry and category of each map element.

## 4 EXPERIMENTS

### 4.1 EXPERIMENTAL SETTINGS

**nuScenes Dataset.** nuScenes (Caesar et al., 2020) is a widely recognized dataset in the field of autonomous driving research, providing 1,000 scenes, each captured over a continuous 20-second interval. Each dataset sample incorporates data from six synchronized RGB cameras and includes detailed pose information. The perception ranges from -15.0m to 15.0m along the X-axis and from -30.0m to 30.0m along the Y-axis. For experimental purposes, the dataset is partitioned into 700 scenes comprising 28,130 samples for training, and 150 scenes containing 6,019 samples for validation.

**Argoverse2 Dataset.** The Argoverse2 dataset (Wilson et al., 2023) contains multimodal data from 1,000 sequences, including high-resolution images from seven ring cameras and two stereo cameras, as well as LiDAR point clouds and map-aligned 6-DoF pose data. All annotations are densely sampled to facilitate the training and evaluation of 3D perception models. Results are reported on the validation set, with a focus on the same three map categories as identified in the nuScenes dataset.

**Evaluation Metric.** In alignment with MapTR (Liao et al., 2022), we adopt the widely-accepted metric of mean Average Precision (mAP) based on the Chamfer distance. The evaluation thresholds are set at 0.5m, 1.0m, and 1.5m. We also utilize the IoU-based metric $mAP^{raster}$ as employed in MapVR (Zhang et al., 2024). Specifically, $AP_{ped}$, $AP_{div}$, and $AP_{bou}$ refer to the average precision for pedestrians, dividers, and boundaries, respectively. Furthermore, real-time performance is measured in terms of frames per second (FPS), and the model's complexity is evaluated through the number of parameters (Params).

**Training Loss.** Our framework builds on MapTRv2 (Liao et al., 2023), which serves as the primary baseline. Therefore, we align our training loss with MapTRv2, including the point loss $\mathcal{L}_{pts}$, the classification loss $\mathcal{L}_{cls}$, the edge direction loss $\mathcal{L}_{dir}$, and the dense prediction loss $\mathcal{L}_{dense}$. To further stabilize training and improve perception, an auxiliary Loss $\mathcal{L}_{aux}$ is designed, which includes instance segmentation loss $\mathcal{L}_{ins\_seg}$ and reference point loss $\mathcal{L}_{ref}$. The instance segmentation loss $\mathcal{L}_{ins\_seg}$ is defined as a combination of cross-entropy loss and dice loss between the predicted instance segmentation map and the ground-truth instance mask, where the instance segmentation map is generated through the dot product of instance-level queries and BEV features. Regarding the reference point loss $\mathcal{L}_{ref}$ is concerned, it accelerates convergence and stabilizes training by supervising reference points of each decoder layer with vectorized map instances. Specifically, Hungarian matching is first performed, followed by the calculation of the L1 distance between each points pair, consistent with $\mathcal{L}_{pts}$. The final loss $\mathcal{L}$ is defined as the weighted sum of the above losses as

$$\mathcal{L} = \beta_1 \mathcal{L}_{pts} + \beta_2 \mathcal{L}_{cls} + \beta_3 \mathcal{L}_{dir} + \beta_4 \mathcal{L}_{dense} + \beta_5 \mathcal{L}_{aux}, \tag{9}$$

where $\beta_i$ represents the weight coefficient of the corresponding loss.

**Implementation Details.** Our model is trained on 8 A100 GPUs with the batchsize of 2, utilizing the AdamW optimizer (Loshchilov et al., 2017) with the learning rate of $4 \times 10^{-4}$. We adopt the ResNet50 (He et al., 2016) as the backbone and employ the LSS transformation (Philion & Fidler, 2020) with a single encoder layer for feature extraction. The one-to-many training strategy (Liao et al., 2023) is used, and the model is trained for 24 epochs on the nuScenes dataset and 6 epochs on the Argoverse2 dataset. The hyperparameters are configured as $N_q = 100$, $N_{rep} = 8$, $N_p = 20$, $\beta_1 = 5$, $\beta_2 = 2$, $\beta_3 = 0.005$, $\beta_4 = 3$, and $\beta_5 = 3$.

## 4.2 COMPARISON WITH STATE-OF-THE-ART METHODS

**Results on nuScenes.** Table 1 presents the results on the nuScenes validation dataset, utilizing multi-view RGB images as input. In comparison to MapTRv2 (Liao et al., 2023), the proposed MGMapNet has reached an mAP of 66.8, exceeding it by 5.3 with a training duration of 24 epochs. After a prolonged training period of 110 epochs, the mAP obtained by MGMapNet is 73.6, which is still significantly higher than 68.7 of MapTRv2 and 72.6 of MapQR (Liu et al., 2024b).

Table 1: Comparison to the state-of-the-art methods on nuScenes val set.

| Method | Epoch | $AP_{ped}$ | $AP_{div}$ | $AP_{bou}$ | mAP | FPS | Params (MB) |
|---|---|---|---|---|---|---|---|
| HDMapNet (Li et al., 2022a) | 30 | 14.4 | 21.7 | 33.0 | 23.0 | - | - |
| BeMapNet (Qiao et al., 2023a) | 30 | 62.3 | 57.7 | 59.4 | 59.8 | 4.3 | - |
| PivotNet (Ding et al., 2023) | 24 | 56.5 | 56.2 | 60.1 | 57.6 | 9.2 | - |
| MapTRv2 (Liao et al., 2023) | 24 | 59.8 | 62.4 | 62.4 | 61.5 | 14.1 | 40.3 |
| MGMap (Liu et al., 2024a) | 24 | 61.8 | 65.0 | 67.5 | 64.8 | 12 | 55.9 |
| MapQR (Liu et al., 2024b) | 24 | **68.0** | 63.4 | 67.7 | 66.4 | 11.9 | 125.3 |
| MGMapNet (Ours) | 24 | 64.7 | **66.1** | **69.4** | **66.8** | 11.7 | 70.1 |
| VectorMapNet (Liu et al., 2023a) | 110 | 42.5 | 51.4 | 44.1 | 46.0 | - | - |
| MapTRv2 (Liao et al., 2023) | 110 | 68.1 | 68.3 | 69.7 | 68.7 | 14.1 | 40.3 |
| MGMap (Liu et al., 2024a) | 110 | 64.4 | 67.6 | 67.7 | 66.5 | 12 | 55.9 |
| MapQR (Liu et al., 2024b) | 110 | **74.4** | 70.1 | 73.2 | 72.6 | 11.9 | 125.3 |
| MGMapNet (Ours) | 110 | 74.3 | **71.8** | **74.8** | **73.6** | 11.7 | 70.1 |

Moreover, the comparison between the proposed MGMapNet and two latest models in terms of IoU-based metrics is performed, with the results shown in Table 2. It is observed that the proposed MGMapNet generally achieves the best IoU-based performances than MapVR and MGMap.

Table 2: Performance comparison on nuScenes val set in terms of IoU-based AP.

| Method | $AP_{ped}^{raster}$ | $AP_{div}^{raster}$ | $AP_{bou}^{raster}$ | $mAP^{raster}$ |
|---|---|---|---|---|
| MapVR (Zhang et al., 2024) [NeurIPS2023] | 46.0 | 39.7 | 29.9 | 38.5 |
| MGMap (Liu et al., 2024a) [CVPR2024] | **54.5** | 42.1 | 37.4 | 44.7 |
| MGMapNet (Ours) | 54.0 | **42.7** | **44.1** | **46.9** |

Moreover, the qualitative results are depicted in Fig. 3, where there are three complex scenarios: daytime vehicles with occlusion, nighttime low-light conditions, and low-light situations with occlusion. In the first case, MGMapNet exhibits more precise coordinate prediction compared to StreamMapNet (Yuan et al., 2024) and preserves all road elements compared to MapTRv2. In the second case of nighttime low-light conditions, MapTRv2 struggles to predict the divider on the right side of the vehicle due to its lack of instance-level perception. While StreamMapNet utilizes instance-level queries and identifies the divider, its overall instance positioning accuracy remains inadequate. In contrast, MGMapNet accurately and completely detects the boundary in these challenging scenarios. In the third case involving nighttime conditions with occlusion, the results of StreamMapNet are nearly unusable; MapTRv2 incorrectly identifies the rear divider as a boundary, revealing its limitation in instance-level perception; by contrast, MGMapNet demonstrates remarkable robustness on accurately identifying the elements.

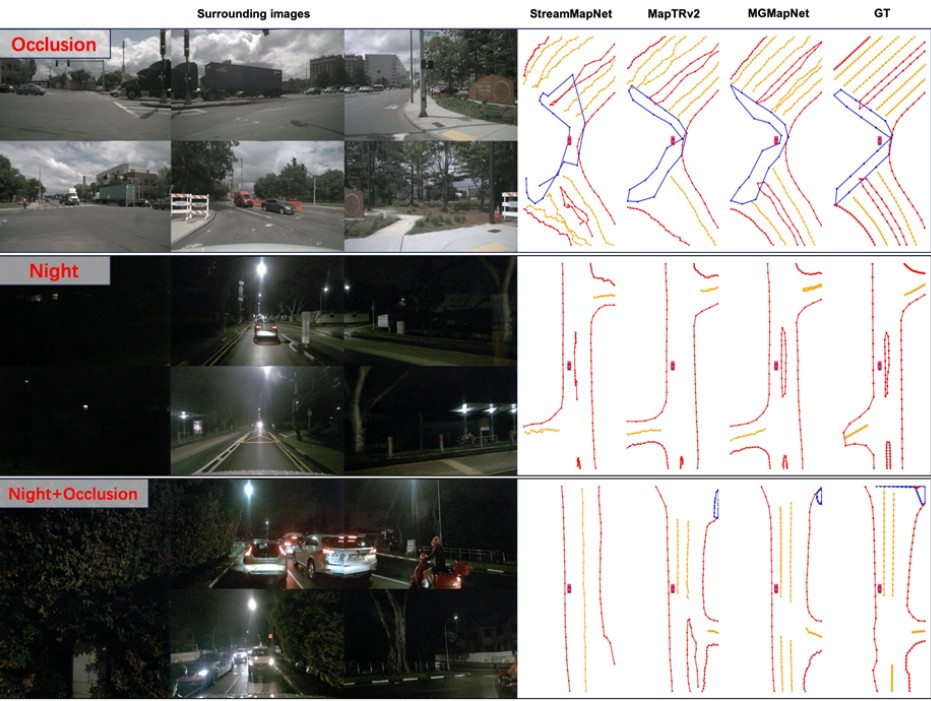

Figure 3: Qualitative visualization on nuScenes val set.

**Results on Argoverse2.** Table 3 presents the results on the Argoverse2 validation dataset for 6 epochs. The Argoverse2 dataset provides two configurations for the representation of points: 2D and 3D point coordinates. We conduct experiments on both configurations, and the proposed MGMapNet outperforms the competing methods.

**Efficiency Comparison.** We conduct a comprehensive efficiency analysis of the proposed MGMapNet and several open-source models. As demonstrated in the last two columns of Table 1, MGMapNet achieves the FPS of 11.7, which is comparable to the latest methods of MapQR and MGMap. The number of model parameters of MGMapNet is 70.1 MB, which is lower than MapQR's 125.3 MB but slightly higher than MGMap's 55.9 MB.

Table 3: Comparison to the state-of-the-art methods on Argoverse2 val set.

| Method | Map dim. | $AP_{ped}$ | $AP_{div}$ | $AP_{bou}$ | mAP |
|---|---|---|---|---|---|
| HDMapNet (Li et al., 2022a) | | 13.1 | 5.7 | 37.6 | 18.8 |
| VectorMapNet (Liu et al., 2023a) | | 38.3 | 36.1 | 39.2 | 37.9 |
| MapTRv2 (Liao et al., 2023) | 2 | 62.9 | 72.1 | 67.1 | 67.4 |
| MapQR (Liu et al., 2024b) | | 64.3 | 72.3 | 68.1 | 68.2 |
| HIMap (Zhou et al., 2024) | | **69.0** | 69.5 | 70.3 | 69.6 |
| MGMapNet (Ours) | | 67.1 | **74.6** | **71.7** | **71.2** |
| VectorMapNet (Liu et al., 2023a) | | 36.5 | 35.0 | 36.2 | 35.8 |
| MapTRv2 (Liao et al., 2023) | | 60.7 | 68.9 | 64.5 | 64.7 |
| MapQR (Liu et al., 2024b) | 3 | 60.1 | 71.2 | 66.2 | 65.9 |
| HIMap (Zhou et al., 2024) | | **66.7** | 68.3 | 70.3 | 68.4 |
| MGMapNet (Ours) | | 64.7 | **72.1** | **70.4** | **69.1** |

## 4.3 ABLATION STUDY

The proposed MGA consists of multi-granularity aggregator and point-instance interaction. In order to reveal the contributions of different modules in the proposed model, ablation study is conducted on the nuScenes dataset with the results shown in Table 4, where the multi-point attention (MPA) from StreamMapNet (Yuan et al., 2024) serves as the baseline. The multi-granularity aggregator is employed as a comparative alternative in the decoder layer, leveraging multi-granularity queries for prediction. We further validate the effectiveness of point-instance interaction by incrementally adding its components. There are several observations from the results. First, replacing MPA with the proposed multi-granularity aggregator improves mAP from 59.6 to 62.7, showing the effectiveness of multi-granularity representations. Second, with multi-granularity aggregator, the separate inclusion of P2P attention and P2I attention increases mAP from 62.7 to 64.8 and 65.0, respectively. This reveals that it is helpful to employ both geometric and category details to enhance queries. Third, by utilizing the complete point-instance interaction module, the model achieves the best mAP of 66.8, highlighting the importance of multi-granularity query interaction in boosting perception.

Table 4: Ablation study of the proposed modules.

| Method | Point-instance Interaction | | mAP |
|---|---|---|---|
| | P2P Attention | P2I Attention | |
| Multi-point Attention (Yuan et al., 2024) | × | × | 59.6 |
| | × | × | 62.7 |
| Multi-granularity Aggregator | ✓ | × | 64.8 |
| | × | ✓ | 65.0 |
| | ✓ | ✓ | 66.8 |

Table 5: Ablation study on the generality of the proposed modules transferred into MapTRv2.

| Experiment | Method | mAP |
|---|---|---|
| | Multi-point Attention | 55.9 |
| (a) | Multi-granularity Attention | 63.6 (+7.7) |
| (b) | + Auxiliary Loss | 64.4 (+0.8) |
| (c) | + Multi-scale BEV Feature | 65.0 (+0.6) |
| (d) | + Position Embeddings | 66.2 (+1.2) |
| (e) | + Increase Query Number | 66.8 (+0.6) |

To verify the generality of the proposed modules, we transfer them into the MapTRv2 (Liao et al., 2023) framework, with the ablation results shown in Table 5. The decoder with the multi-point attention is used as the baseline, and we design the following experiments: (a) replace multi-point attention with MGA; (b) add the auxiliary loss; (c) utilize multi-scale BEV features; (d) incorporate

position embeddings into the multi-granularity aggregator; and (e) increase the number of queries. The experimental results demonstrate the effectiveness of these designed techniques.

## 5 CONCLUSION AND DISCUSSION

In this work, we propose a multi-granularity map network for end-to-end vectorized HD map construction. Specifically, the proposed multi-granularity attention leverages coarse-grained instance queries to represent fine-grained point queries. Point-instance interaction further captures category and geometric information by facilitating interaction between features of different granularities. The proposed method enhances map construction performance by employing multi-granularity queries to perceive map instance categories and polyline distributions. Extensive experiments are conducted on two benchmark datasets to verify the superiority of the proposed method compared with other state-of-the-art approaches. In the future, it is desired to extend the current work by considering the following two aspects: (1) exploring temporal methods to incorporate additional prior information, and (2) investigating more efficient representation mechanisms to enhance model efficiency.

### ACKNOWLEDGMENTS

This work was supported in part by National Natural Science Foundation of China under Grant 62371343 and Shanghai Municipal Science and Technology Major Project (No. 2021SHZDZX0100).

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
