# OpenReview forum: "MGMapNet: Multi-Granularity Representation Learning for End-to-End Vectorized HD Map Construction"
_ICLR.cc/2025/Conference — ICLR 2025 Poster_

### Official Review · Reviewer_pXZj · 2024-11-01

**Soundness:** 3
**Presentation:** 3
**Contribution:** 3
**Rating:** 6
**Confidence:** 2

**Summary:**

The paper presents MGMapNet, a framework designed for end-to-end vectorized High-Definition map construction. MGMapNet introduces a multi-granularity representation that integrates both instance-level and point-level queries to effectively capture both category and geometric information of road elements. The proposed Multi-Granularity Aggregator and Point Instance Interaction modules allow information exchange between the two granularities, resulting in enhanced prediction accuracy. Experimental results demonstrate state-of-the-art performance on benchmark datasets such as nuScenes and Argoverse2, surpassing various baseline methods.

**Strengths:**

S1. The paper introduces a method that combines both coarse-grained instance-level and fine-grained point-level queries, effectively capturing both global category information and local geometric details of map elements.

S2. The design of the Multi-Granularity Aggregator and Point Instance Interaction modules facilitates efficient and effective information sharing between instance-level and point-level queries.

S3. The proposed MGMapNet framework outperforms several baseline models, achieving the state-of-the-art performance in HD map construction.

**Weaknesses:**

W1. The paper’s description can be overwhelming for readers who are not deeply familiar with the HD map construction topic (e.g., me). For example, it lacks a formal problem formulation, which would help in grounding the research context. Additionally, the method's explanation is a bit difficult to follow.

W2. The paper could be strengthened by providing a detailed analysis of the time and space complexity of MGMapNet compared to baseline models. Given that efficiency is a key motivation, understanding how MGMapNet performs in terms of computational and memory resources would be beneficial.

W3. It is not clear why the training epochs are set to have multiple values for various models, and why the long training schedule leads to fair comparison.

**Questions:**

Please clarify the comments for W1-W3.

---

> ### Author Response · Authors · 2024-11-19
> **Response to Reviewer pXZj**
>
> We thank the reviewer for the supportive comments. The detailed response to each point is as follows.
>
> > **W1.The paper’s description can be overwhelming for readers who are not deeply familiar with the HD map construction topic (e.g., me).**
>
> - Thank you for your valuable suggestion.
> - The map reconstruction task primarily involves obtaining unified Bird’s-Eye View (BEV) features $\mathbf{F}_{bev}\in\mathbb{R}^{C\times H\times W} $( $C, H, W$ represent the feature channels, height, and width of the BEV feature) from surround-view cameras images $I$.
> Subsequently, a DETR-like decoder is employed to perceive and vectorize map elements.
> Each vectorized map element $\mathbf{P}$,
> comprises a category (such as pedestrians, dividers, and boundaries) and a series of consecutive vector coordinate points $\\{v_i\\}\_{i=0}^{N_p-1}$, where $N_p$ is the number of points and $v_i$ is the coordinate of the $i$-th point. This vectorized representation allows for a more precise depiction of map elements, resulting in high-precision map polylines.
>
> - The principal challenge of this problem is to capture precise local coordinates while simultaneously learning and modelling each instance.
> To address this challenge, our model introduces a multi-granularity representation mechanism that facilitates the simultaneous modeling of entire instances and their intricate points, thereby improving the performance of High-Definition vectorized map representations.
>
> > **W2.The paper could be strengthened by providing a detailed analysis of the time and space complexity of MGMapNet compared to baseline models.**
>
> |                   | MapTR [ICLR2023] | MapTRv2 [IJCV2024] | MGMap [CVPR2024] | MapQR [ECCV2024] | MGMapNet    |
> |:-----------------:|:-----------------:|:-------------------:|:----------------:|:-----------------:|:-----------:|
> | **FPS**           | **16.9**          | 14.1                | 12               | 11.9              | 11.7        |
> | **GPU mem. (MB)** | **2314**          | 2656                | 2402             | 2648              | 2790        |
> | **Params. (MB)**   | **35.9**          | 40.3                | 55.9             | 125.3             | 70.1        |
> | **NuScenes (mAP)**  | 50.3              | 61.5                | 64.8             | 66.4              | **66.8**    |
> | **Argoverse2 (mAP)**| 58                | 67.4                | -                | 68.2              | **71.2**    |
>
> - We agree that analyzing the computational and memory resources is essential for assessing efficiency.
>
> - In Table above, we present a comprehensive comparison of the latest models alongside the primary baseline, detailing GPU memory usage, FPS, parameter counts, and performance.
> Time and space complexity can be derived from FPS and GPU memory comparisons.
>
>
>   - **GPU mem. comparison.** The memory usage (MB) of MapTR, MapTRv2, MGMap, MapQR, and MGMapNet are 2314, 2656, 2402, 2648, and 2790 respectively. Our MGMapNet has a slight increase in memory usage compared to other methods, which is understandable given we retained two types of queries for different output regressions and classifications.
>
>   - **FPS comparison.** MGMapNet, MapQR, and MGMap show similar performance with FPS scores of 11.7, 11.9, and 12, respectively. Although slightly slower than MapTRv2, MGMapNet's inference time complexity is similar to that of the latest methods.
>
>   - **Params comparison.** The parameters (MB) of MGMapNet, MapQR, and MGMap are 70.1, 125.3, and 55.9, respectively. Even though MGMapNet has a slightly higher parameter count due to its Multi-Granularity query design and Point Instance Interaction, it still outperforms and has fewer parameters than MapQR’s 125.3MB. We believe there’s substantial room for optimization in MGMapNet.
>
> - In an overall efficiency analysis, our MGMapNet, thanks to its multi-granularity representation, achieves better performance while maintaining similar parameters, speed, and memory usage compared to the latest methods. The limitations of our method in terms of speed have been mentioned, but we believe there is a significant room for optimization. Therefore, MGMapNet remains a competitive model.
>
> > **W3.It is not clear why the training epochs are set to have multiple values for various models, and why the long training schedule leads to fair comparison.**
>
> - Previous comparative studies generally employed two distinct training epoch configurations, conducting long epochs to maintain consistency with prior methods and ensure a fair comparison.
>
> - On one hand, shorter training epochs might emphasize the models’ short-term overall performance and may result in some models not fully converging. On the other hand, longer training epochs ensure that models generally reach a converged state. This dual setup allows us to evaluate the performance of each model more objectively and comprehensively.
>
> Thanks again and we are happy to take any questions / further discussions.

---

> ### Author Response · Authors · 2024-11-25
> **Kind Reminder to Reviewer pXZj for the Feedback on Our Rebuttal**
>
> Dear Reviewer pXZj,
>
> Thank you sincerely for your thoughtful review and valuable comments. Although we may not be from exactly the same field, we deeply appreciate your insights and the opportunity to engage with your feedback during the discussion period.
>
> Your comments are highly insightful, and we believe they offer meaningful guidance to further strengthen our work. In our rebuttal, we have carefully addressed each of your concerns with detailed responses. Specifically, we have:
>
> - Added an experimental analysis on efficiency in the supplementary materials and included additional challenges and contributions in Sections 1 and 3 in the latest revised version we uploaded.
>
> - Provided explanations regarding the experimental setup and problem description in the rebuttal.
>
> We hope our responses have addressed some of your concerns. If there are any additional questions or further clarifications required, please do not hesitate to let us know. We would be more than happy to provide further details.
>
> Thank you once again for your time and thoughtful review.
>
> Best regards,
>
> Authors of #5440

---

> > ### Comment · Reviewer_pXZj · 2024-11-26
> >
> > Thanks for your response. I have no more questions and will maintain my score.

---

> > > ### Author Response · Authors · 2024-11-26
> > > **Thanks for your feedback**
> > >
> > > Dear Reviewer pXZj,
> > >
> > > Thank you for your feedback! We sincerely appreciate your constructive review and valuable suggestions, which have been incredibly helpful in improving our work.
> > >
> > > Best regards,
> > >
> > > Authors of #5440

---

### Official Review · Reviewer_cU8c · 2024-11-03

**Soundness:** 3
**Presentation:** 2
**Contribution:** 2
**Rating:** 6
**Confidence:** 3

**Summary:**

The paper introduces MGMapNet, designed to effectively model map elements through a multi-granularity representation by integrating both coarse-grained instance-level and fine-grained point-level queries to enhance map modeling. The framework employs a Multi-Granularity Aggregator. Besides, there is a Point Instance Interaction module, which facilitates the exchange of information between the instance-level and point-level queries, thereby improving the overall modeling capability of the network.

**Strengths:**

1.	The problem studied in the paper is very important in practice and find applications in real world.
2.	The paper is clearly written and easy to follow.
3.	Experiments are conducted to verify the performance of the proposed method.

**Weaknesses:**

1.	The challenges and contributions of the proposed techniques require further elaboration. What are the specific challenges to design these techniques in section 3?
2.	The encoders and decoders are mostly MLP-based. It is difficult to understand the logic, rationale and difficulty to apply the techniques.
3.	Some evaluation metrics in experiments are not explained, e.g. AP_ped and AP_div, and AP_bou in table 1.
4.	How are the proposed techniques related to High-Definition?
5.	Quality of figures and tables can be improved. For example, Table 4 has too big font size.

**Questions:**

1.	Is it possible to consider instance-2-instance attention? Why not compare this?

---

> ### Author Response · Authors · 2024-11-19
> **Response to Reviewer cU8c Part I (Part I of II)**
>
> We thank the reviewer for the supportive comments. The detailed response to each point is as follows.
>
> > **W1.The challenges and contributions of the proposed techniques require further elaboration. What are the specific challenges to design these techniques in section 3?**
>
> - We apologize for the unclear expression.
> - The main challenge is to balance the trade-off between detail and overview representation. Existing state-of-the-art (SOTA) methods, such as MapTR and MapTRv2, utilize point-level queries to characterize map elements, whereas StreamMapNet employs instance-level queries. Each granularity of queries has its own advantages and disadvantages:
>
>    - **Instance-level queries** excel at capturing the overall category information of road elements but may struggle to accurately represent geometric details, especially for irregular or elongated map elements.
>
>    - **Point-level queries** can provide rich, detailed information, but can only represent instances by combining multiple point-level queries, lacking an overarching description of map elements.
>
> - The balance between detail and overview remains a major challenge in current research, and existing methods do not adequately address this issue.
>
> - Our contribution is the simultaneous acquisition of both detailed and comprehensive instance features. The multi-granularity mechanism is designed to solve this problem by effectively maintaining and updating queries of different granularities. MGMapNet overcomes the limitations of existing methods, achieving a balance between detailed and overall feature representations.
> - We will elaborate it further in the revised version.
>
> > **W2.The encoders and decoders are mostly MLP-based. It is difficult to understand the logic, rationale and difficulty to apply the techniques.**
>
> - We did not fully understand this statement. If we are not mistaken, the reviewer may be referring to positional encoding.
> - The MLP-based structure in our decoder, as detailed in Equations 2 and 5, is primarily utilized to generate positional encodings. While many existing methods employ sinusoidal encodings for this purpose, our empirical observations indicate that sinusoidal encodings may exhibit inferior generalization capabilities and reduced performance compared to adaptive encodings implemented via MLPs.
> In Equation 8, employing an MLP to ensure the adequate aggregation of queries at two levels of granularity is considered a relatively reasonable approach.
>
> > **W3.Some evaluation metrics in experiments are not explained, e.g. $AP_{ped}$ and $AP_{div}$, and $AP_{bou}$ in table 1.**
>
> - Thank you for pointing out the omission of explanations for some evaluation metrics in our experiments. We apologize for this oversight. Specifically, $AP_{ped}$ and $AP_{div}$, and $AP_{bou}$ refer to the Average Precision for pedestrians, dividers, and boundaries, respectively.
> - We will include these clarifications in the revised manuscript to enhance clarity and comprehensiveness.
>
> > **W4.How are the proposed techniques related to High-Definition?**
>
> - Traditional autonomous driving systems rely on high-precision maps created through offline annotation. In contrast, this paper primarily addresses sensor-based high-precision map reconstruction.
> The construction of high-definition maps using purely visual methods has become an increasingly challenging endeavour since MapTR.
>
> - Vectorized HD map construction requires a higher level of precision, as map elements are represented using vectorized points to accurately depict features such as pedestrians, dividers, and boundaries. Our proposed MGMapNet, along with recent papers in the field, are specifically designed to enhance the accuracy and precision of map reconstruction.
>
> > **W5.Quality of figures and tables can be improved. For example, Table 4 has too big font size.**
>
> - Thank you for your suggestions. The fonts in the figures were an oversight in our writing, and we will correct them in the revised version.

---

> ### Author Response · Authors · 2024-11-19
> **Response to Reviewer cU8c Part II (Part II of II)**
>
> > **Q1:Is it possible to consider instance-2-instance attention? Why not compare this?**
>
> - Thank you for your valuable suggestion.
>
> - In fact, MGMapNet employs instance-to-instance attention. As shown in Figure 2 of the paper, the Self Attention preceding Multi-Granularity Attention follows this structure and facilitates interactions among instances, aligning with the default configuration.
> Consequently, incorporating instance-to-instance attention within MGA is unnecessary, as inter-instance interactions are already implemented prior to input through the Self Attention module.
>
> - Multi-Granularity Attention begins by generating queries at multiple granularities and optimizing them through point-instance interaction. The point-to-instance attention enables corresponding instances to more effectively aggregate point features.
> Additionally, point-to-point attention not only within the current layer but also emphasises point features from the previous layer of the decoder.
> This integrated approach facilitates a coarse-to-fine refinement process for high-precision point queries, enhancing the model's ability to capture detailed spatial information.
> Since the point query in MGA is optimized from coarse to fine, and the instance query itself is a coarse-grained query that has already been implemented in Self Attention, we did not include instance-to-instance attention within MGA.
>
> - In future work, integrating Self Attention within MGA to explore more comprehensive multi-granularity interactions constitutes a potential research direction.
>
> Thanks again and we are happy to take any questions / further discussions.

---

> ### Author Response · Authors · 2024-11-25
> **Kind Reminder to Reviewer cU8c for the Feedback on Our Rebuttal**
>
> Dear Reviewer cU8c,
>
> Thanks for your thoughtful review and valuable comments. During the discussion period, we also want to get some feedback from you.
>
> Actually, your comments are particularly insightful, and we believe they will help strengthen our work significantly. In our rebuttal, we have carefully addressed each of your concerns with detailed responses. Specifically, we have included
> - Included detailed explanations of the technological challenges and contributions, descriptions of the evaluation metrics used in the experiments, improved chart formats for better readability, and reorganized Figure 2 for enhanced clarity and aesthetics in the revised version we have uploaded.
> - Provided explanations of the MLP-based model structure, proposed techniques related to High-Definition, and addressed the issue of instance-to-instance attention during the rebuttal phase.
>
> We would sincerely appreciate it if we could get some feedback from you regarding the above concerns. If you have any further questions or require additional clarifications, please do not hesitate to let us know.
>
> Thank you for your time and consideration.
>
> Best regards,
>
> Authors of #5440

---

> > ### Comment · Reviewer_cU8c · 2024-11-26
> > **Thanks for the feedback.**
> >
> > The feedback has addressed most of my concerns. I have updated the rating. Thanks.

---

> > > ### Author Response · Authors · 2024-11-26
> > > **Thanks for your feedback**
> > >
> > > Dear Reviewer cU8c,
> > >
> > > Thank you for your feedback! We greatly appreciate the constructive reviews and valuable suggestions to enhance our work.
> > >
> > > Best regards,
> > >
> > > Authors of #5440

---

### Official Review · Reviewer_hewp · 2024-11-04

**Soundness:** 4
**Presentation:** 4
**Contribution:** 4
**Rating:** 6
**Confidence:** 3

**Summary:**

This paper presents MGMapNet, a multi-granularity map network for end-to-end vectorized HD map construction based on multi-scale bird’s eye view (BEV) images. Evaluations on four datasets show the effectiveness of MGMapNet over multiple baseline models.

**Strengths:**

1. The contributions are highlighted. The novel contributions compared with previous approaches are also discussed properly.
2. Both quantitative and qualitative results are shown and discussed. Ablation studies are conducted in a meaningful way.

**Weaknesses:**

1. The citations of the whole paper are wrong. It should be \citep{} instead of \cite{}.

2. From Figure 1 and 3. we can see the advantages of MGMapNet over other models. However, I can still see that the extracted lanes by MGMapNet are sometimes zigzagged while the ground truth lines are straight lines. I wonder whether you can add some regularity or loss terms to avoid this. Maybe for those straight lines, resample their vertices along the straight lines every times during model training so that the model learns the linear feature instead of individual point locations?

3. Can you describe each loss term in detail? Do you use the distance from each point to the ground truth points as the loss for L_{pts}?

**Questions:**

See the weakness.

---

> ### Author Response · Authors · 2024-11-19
> **Response to Reviewer hewp Part I (Part I of II)**
>
> We thank the reviewer for the supportive comments. The detailed response to each point is as follows.
>
> > **W1. The citations of the whole paper are wrong.**
>
> - This was an oversight in our writing. In the revised version, we will correct all the citations to ensure they follow the appropriate format.
>
> > **W2. From Figure 1 and 3. we can see the advantages of MGMapNet over other models. However, I can still see that the extracted lanes by MGMapNet are sometimes zigzagged while the ground truth lines are straight lines. I wonder whether you can add some regularity or loss terms to avoid this.**
>
> - Thank you very much for your insightful suggestions. Geometric properties have always been a focus of research. PivotNet (CVPR 2023) uses the inherent properties of polylines to introduce the concepts of pivot points and collinear points.
>
> - Your suggestion to put more emphasis on the positions of these pivot points is indeed reasonable. However, in our current version, we have primarily developed more advanced representations within the MapTRv2 framework.
>
> - Employing instance segmentation as auxiliary supervision facilitates the optimization of specific zigzagged points.
> Precise rasterized instance masks for each instance query can partially reduce point instability. In addition, the use of auxiliary instance segmentation loss, also adopted in our method, further prevents the generation of anomalous points within map elements.
> However, as shown in Figure 3, such corner cases cannot be completely avoided.
> - In future work, we aim to further optimize our approach by exploiting geometric relationships.

---

> ### Author Response · Authors · 2024-11-19
> **Response to Reviewer hewp Part II (Part II of II)**
>
> > **W3.  Do you use the distance from each point to the ground truth points as the loss for $\mathcal{L}_{pts}$?Can you describe each loss term in detail?**
>
> - Yes, we do, and we apologize for the unclear descriptions of each loss term.
>  As you mentioned, the $\mathcal{L}_{pts}$ is the loss calculated between each predicted point and the corresponding ground truth points.
>
> - Firstly, we find an optimal instance-level label assignment between predicted map elements and ground truth map elements using the Hungarian algorithm at the instance-level.
> Secondly, the predicted points are paired with the ground truth points using the Hungarian algorithm at the point level, establishing a one-to-one point correspondence between
> $\mathbf{p}\_i^{pred}$ and $\mathbf{p}\_i^{gt}$ ($\mathbf{P}\_i$ is the $i$-th coordinate points of instance). Following this matching, a point-wise $\mathcal{L}\_{pts}$ loss is applied to optimize the predictions. This loss calculation approach follows the methodology outlined in MapTR and MapTRv2.
> The $\mathcal{L}\_{pts}$ is formulated as:
> $$ \mathcal{L}_{pts} = \frac{1}{N_p}\sum\_{i=1}^{N_p} \\| \mathbf{p}_i^{pred} - \mathbf{p}_i^{gt} \\|_1 $$
> where $\mathbf{p}_i^{pred}$ and $\mathbf{p}_i^{gt}$ are the predicted and ground truth positions of point $i$, respectively, and $N_p$ is the number of points.
>
> - The $\mathcal{L}\_{pts}$, $\mathcal{L}\_{cls}$, $\mathcal{L}\_{dir}$ and $\mathcal{L}\_{dense}$ loss align with MapTRv2.
> Additionally, we introduce auxiliary losses, which comprise two components: the instance segmentation loss $\mathcal{L}\_{insseg}$ and the reference point loss $\mathcal{L}\_{ref}$.
>
> - The instance segmentation loss, denoted as $\mathcal{L}\_{insseg}$, not only segments BEV features but also retrieves more precise instance localization information for each individual query.
> First, we compute the instance segmentation masks $M_{i}^{pred} \in \mathbb{R}^{H\times W \times N_q }$ by performing dot product operations between the updated instance-level queries $\mathbf{Q}\_{ins}\in\mathbb{R}^{N_q \times C}$ and the BEV features $\mathbf{F}\in\mathbb{R}^{H\times W\times C}$.
> Subsequently, we utilize the indices of positive samples obtained through the Hungarian algorithm to retrieve their corresponding masks $M_{pos}^{pred}$ and ground truths $M_{pos}^{gt}$.
> For each positive sample instance mask $M_{pos}^{pred}\in\mathbb{R}^{H\times W\times N_{pos}}$ ($N_{pos}$ is the total number of positive query), we separately compute the segmentation loss by employing both Binary Cross-Entropy loss $\mathcal{L}\_{bce}$ and Dice loss $\mathcal{L}\_{Dice}$.
> The process of generating the $M_{pos}$ is formulated as:
> $$M^{pred}=\mathbf{F} \cdot \mathbf{Q}\_{ins}^{T},$$
> where $\cdot$ denote dot product operations and the $\mathcal{L}\_{insseg}$ is formulated as:
> $$\mathcal{L}\_{insseg} = \frac{1}{N\_{pos}} \sum\_{i=1}^{N_{pos}} (\mathcal{L}\_{dice}(M\_{pos,i}^{pred},M\_{pos,i}^{gt})+\mathcal{L}\_{bce}(M\_{pos,i}^{pred},M\_{pos,i}^{gt})), $$
> where $M_i$ denote the $i$-th positive instance mask..
>
> - Additionally, the reference point loss $\mathcal{L}\_{ref}$ provides auxiliary supervision for reference points during each iteration of the decoder.
> Similar to the $\mathcal{L}\_{pts}$ loss, The $\mathcal{L}\_{ref}$ is computed by applying the $\mathcal{L}\_{pts}$ loss to the reference points $\textbf{RF}$ and ground truth points $\textbf{P}^{gt}$ at each layer. This ensures that each sampling point achieves a more reasonable and accurate distribution.
>
> - We will provide further elaboration on these details in the revised version.
> Thanks again and we are happy to take any questions / further discussions.

---

> ### Author Response · Authors · 2024-11-25
> **Kind Reminder to Reviewer hewp for the Feedback on Our Rebuttal**
>
> Dear Reviewer hewp,
>
> Thank you for your thoughtful review and valuable feedback. We sincerely appreciate the time and effort you have dedicated to evaluating our work.
> Your comments are highly insightful, and we believe they have provided us with an excellent opportunity to further refine and improve our paper. In our rebuttal, we have carefully addressed all of your concerns and provided detailed responses.
> Specifically, we have:
>
> - In the latest revised version we uploaded, we corrected the citation format throughout the paper and added a detailed description of the loss function in the appendix.
> - Regarding weakness 2 concerning the zigzagged corner case, we provided an explanation and offered some attempts to address it.
>
> We would be truly grateful to receive your feedback on the points we have addressed. If you have any further questions or require additional clarifications, please do not hesitate to let us know.
>
> Thank you for your time and consideration.
>
> Best regards,
>
> Authors of #5440

---

> > ### Comment · Reviewer_hewp · 2024-11-25
> > **Response**
> >
> > The authors' response address my concern and I will keep my score as 6.

---

> > > ### Author Response · Authors · 2024-11-26
> > > **Thanks for your feedback**
> > >
> > > Dear Reviewer hewp,
> > >
> > > Thank you for your feedback. We greatly appreciate the valuable suggestions to enhance our work.
> > >
> > > Best regards,
> > >
> > > Authors of #5440

---

### Author Response · Authors · 2024-11-24
**Official Comment by Authors**

We thank the reviewers for their insightful comments and for the positive feedback provided on the paper.
We have uploaded a new version of the manuscript, incorporating reviewer suggestions and addressing the points raised, marking changes in blue text. In particular, we have included:

- Corrected the citation format throughout the paper. (Reviewer hewp)
- Added a detailed description of the loss function in the appendix. (Reviewer hewp)
- Added a detailed description of the technological challenges and our contributions. (Reviewer cU8c)
- Added a description of some evaluation metrics in the experiment. (Reviewer cU8c)
- Modified the chart formats for improved legibility. (Reviewer cU8c)
- Figure 2 has been reorganized for improved clarity and aesthetics. (Reviewer cU8c)
- Included more detailed efficiency comparisons in the appendix. (Reviewer pXZj)

We believe these revisions have strengthened the paper and look forward to further feedback. Below, we offer specific responses to the individual comments from each reviewer.

---

### Meta-Review · Area_Chair_kAhD · 2024-12-16

**Metareview:**

The reviewers agree that the problem studied is important in practice, the paper is clearly written and easy to follow. and the experiments are extensively conducted to verify the effectiveness of the model. The reviewers also raised some issues on the unclear explanation on the motivations and contributions of the paper. The writing and more explanations on the figures should be also improved. Some typos should be careful corrected in the final version.

**Additional Comments On Reviewer Discussion:**

The authors have provided rebuttals. During the discussion, some reviewers think some of their concerns are addressed and thus they would like to raise their scores.

---

### Decision · Program_Chairs · 2025-01-22

Accept (Poster)